# An Assessment of Balance through Posturography in Healthy about Women: An Observational Study

**DOI:** 10.3390/s21227684

**Published:** 2021-11-19

**Authors:** Elena Escamilla-Martínez, Ana Gómez-Maldonado, Beatriz Gómez-Martín, Aurora Castro-Méndez, Juan Antonio Díaz-Mancha, Lourdes María Fernández-Seguín

**Affiliations:** 1CPUEX, Centro Universitario de Plasencia, University of Extremadura, Avda. Virgen del Puerto 2, 10600 Plasencia, Spain; escaelen@unex.es (E.E.-M.); anagmpodologa@gmail.com (A.G.-M.); bgm@unex.es (B.G.-M.); 2Podiatry Department, University of Seville, C/Avicena s/n, 41009 Seville, Spain; auroracastro@us.es; 3Physiotherapy Department, University of Seville, C/Avicena s/n, 41009 Seville, Spain; jdm@us.es

**Keywords:** balance, falls, posturography, women, aging, NedSVE/IBV^®^ platform

## Abstract

The incidence of falls in adults constitutes a public health problem, and the alteration in balance is the most important factor. It is necessary to evaluate this through objective tools in order to quantify alterations and prevent falls. This study aims to determine the existence of alteration of balance and the influence of age in a population of healthy women. Static posturography was performed on 49 healthy adult women with no history of falls in four different situations using the Romberg test with the NedSVE/IBV^®^ platform. The variables studied were the body sway area and the anteroposterior and mediolateral displacements. The situation of maximum instability occurred in RGC (*p =* 0.001), with a significant increase in anteroposterior oscillations regarding the ML (*p <* 0.001), with no correlation to age. Age alone does not influence the balance in the sample studied, other factors must come together to alter it. The joint cancellation of visual and somatosensory afferents could facilitate the appearance of falls, given that it is a situation of maximum instability. Proprioceptive training is interesting as a preventive strategy for falls.

## 1. Introduction

Falls in older adults, whose fall incidence increases with age [1], constitute a major public health problem due to morbidity, mortality and the economic and emotional costs for the individual, their family and society [2,3,4,5]. The loss of independence due to immobility is the worst consequence suffered those who experience falls [6].

The etiology of falls is multifactorial, with impaired balance being the most important cause of falls [7,8,9]. Balance is achieved thanks to the interaction between sensory receptors located in the vestibular, visual and somatosensory systems, the Central Nervous System and the musculoskeletal reflex arcs [9].

It is necessary to study each of the three systems for a correct assessment and diagnosis of balance [10,11]. It is possible to train these systems partially or to use all three systems globally, with different methodologies reflected in the scientific field, therefore obtaining an improvement in postural control [12,13,14,15].

Static posturography provides sensitive information about the oscillations of the center of gravity that are generated in an upright static stance [16]. It is a fast, non-invasive and useful tool for screening for possible balance abnormalities [17].

We propose a study of balance via static posturography in healthy older women with no previous history of falls to determine: (1) whether there is an alteration in balance and in which component it occurs (visual (VIS), vestibular (VES) or somatosensory (SOM)); (2) whether balance changes with age; and (3) which of the variables studied (AP displacement, ML displacement or body sway area) undergoes the most changes.

## 2. Materials and Methods

### 2.1. Design

A longitudinal cohort study was carried out following the ethical principles set out in the Declaration of Helsinki on studies with human subjects. The Bioethics Commission of the University of Extremadura approved the study (code project 16-3-20). All participants signed an informed consent as dictated by the Declaration of Helsinki.

This study was performed and reported according to the Strengthening the Reporting of Observational Studies in Epidemiology Criteria (STROBE) [18].

### 2.2. Participants

The initial sample was composed of 53 women, belonging to the Senior Citizens University of the University of Extremadura (Spain). We chose women over 50 years of age, since the highest risk of falls has been demonstrated in older adult women starting from that age [19,20]. The inclusion criteria were: not to present a personal history related to vestibular, peripheral or central pathology; not to present cranioencephalic, osteoarticular or muscular traumas that prevented a correct upright stance and/or ambulation; not to be taking medication that affected the posture such as sedatives of the vestibular apparatus, antidepressants or anxiolytics; and not to exhibit dizziness or vertigo at the time of study. As an exclusion criterion, none of the women should present percentage values below normal according to the indicators given by the platform.

Of the total sample, 4 cases were ruled out for exhibiting pathological values in the somatosensory system once the average values of the Dinascan/IBV force platform were identified. Finally, a sample of 49 women, with an average age of 62.84 ± 7.76 years (minimum 50, maximum 78), a BMI of 26.08 ± 3.57, and with normal balance assessment percentages (SOM = 97%, VIS = 100%, VES = 100%), freely agreed to participate in the study.

### 2.3. Data Collection Protocol

Postural stability in an upright stance was quantified by static posturography using the Dinascan/IBV force platform (600 × 370 mm of active area, 100 mm in height and 25 kg in weight) using the NedSVE/IBV^®^ system, (Valencia, Spain) (Figure 1).

Participants were asked to find the most stable position while barefoot with their arms relaxed on either side of the body, with their heels together and toes apart at a 30° angle. Four static conditions with increasing difficulty were measured, each with a duration of 30 s: Romberg test with eyes open (ROA), Romberg test with eyes closed (ROC), Romberg test with eyes open placing a foam rubber pad under the feet (RGA) and Romberg test with eyes closed and foam rubber pad under the feet (RGC). The foam rubber pad had a thickness of 9 cm and a density of 56.7 kg/m^3^, the penetration resistance was 25% (246 N), and the size was the same as that of the platform (Figure 2).

The balance assessment is based on the comparison of the parameters that best discriminate the pathology of the general population with those obtained from patterns of normality segmented by age (database of the Institute of Biomechanics of Valencia). Ratings are displayed in % so that results other than 100% reflect discrepancy with respect to normal values. It is considered pathological when the results are less than 95%, indicating that these subjects are more likely to present an alteration in the studied system.

The variables studied in the tests were:−Body sway area (mm^2^): approximate area in which the subject’s balance takes place. To obtain this calculation, the software application determines an ellipse that encompasses a group of points that represent the subject’s trajectory during the duration of the test.−Maximum anteroposterior (AP) and mediolateral (ML) displacement (mm): these represent the furthest point reached by the centers of pressure in the anteroposterior and mediolateral axes during the recording time.

### 2.4. Statistical Analysis

The statistical analysis was performed using the SPSS software package version 26.0 (SPSS Inc., Chicago, IL, USA). The normality distribution of the variables was determined with the Shapiro–Wilk test. A univariate analysis of variance (ANOVA) was conducted to assess the changes in the study variables, using the Bonferroni adjustment for post hoc pairwise comparisons.

Paired differences between AP and ML displacement were compared using the Wilcoxon test. To determinate the influence of age on the study measures, a Pearson (r) or Spearman correlation (rho) analysis was performed according to distribution of the variables. The significance level was established as *p* < 0.05.

## 3. Results

A total of 59 women with a mean age of 62.84 ± 7.77 years in ranges of 50–55 (*n* = 7), 55–60 (*n* = 8), 60–65 (*n* = 13), 65–70 (*n* = 12), 70–75 (*n* = 4) and 75–80 (*n* = 5) were recruited. Anthropometric parameters of women are shown in Table 1.

In the ANOVA, we detected significant interaction based on the test performed for the three variables analyzed. Table 2 lists its average values.

In the pairwise comparison, a statistically significant difference was observed in the Romberg test with closed eyes and foam rubber pad (RGC) between the body sway area (F = 33.21; *p* = 0.01; η^2^ = 1), ML displacement (F = 19.15; *p* = 0.01; η^2^ = 1) and AP displacement (F = 32.44 *p* = 0.01; η^2^ = 1) variables.

When comparing the average values of the variables AP displacement and ML displacement in each of the four measured situations (ROA, ROC, RGA, RGC), we observed that the change detected is statistically significant (*p* < 0.001).

Table 3 shows the influence of age on each variable. We observe that there is no significant correlation.

## 4. Discussion

Static posturography helped us detect balance anomalies as the main preventive method against falls.

Of the 53 women studied, only four had alterations in the somatosensory component, who were subsequently eliminated from the study so as not to interfere with the data on the rest of the women. Of the remaining 49, none presented values below 100% in the assessment of the visual and vestibular system. However, although the assessment of the somatosensory system was within the range of normality, in 16 women this value ranged between 95–99%. We could therefore deduce that, without exhibiting pathology, the greatest adaptations occur in this system.

According to the results obtained, where the changes in the variables are statistically significant in the RGC (*p* = 0.001), it can be assumed that the greatest preventive intervention of falls in older people should focus precisely on the improvement of somatosensory or proprioceptive perception.

Although the etiology of falls is complex, there are authors who postulate that falls mostly originate from alteration of the proprioceptive system, making it the most important causal factor. The modification of this system through its main elements—the eyes (convergence) and feet (varus/valgus)—seems essential to obtain postural balance [4]. In this sense, proprioceptive training has been shown to induce positive changes in postural balance and stability [21,22].

It is possible that the use of inappropriate footwear is another factor that influences the results, given that it causes a structural change in the foot and, with it, a modification in proprioception during static loading of the foot [23], although we have not studied the type of footwear that the women usually used.

According to the results obtained, the age in the sample of women studied does not seem to influence the appearance of alterations in balance. This differs from other studies where balance disturbances due to visual, auditory and somatosensory alterations are directly related to age [6,24,25]. However, these studies were carried out in women with an average age greater than those in our sample. In contrast, other research determines that age, by itself, is not a factor that affects the alteration in balance, but it is when combined with different factors that appear especially around the age of 65. Considering that the average age of our sample was established at 62.84 ± 7.76 years, it is possible that these factors were not yet present [26].

The increase in oscillations in older people implies greater instability and, consequently, a greater risk of falls [27]. Our results show greater oscillations in the anteroposterior direction (*p* < 0.001), which is consistent with studies also carried out with older women or with elderly people of both sexes. Since the projection of our center of gravity passes in front of the third sacral vertebra, anteroposterior oscillations in search of postural stability are normal in humans [28,29]. The increase in mediolateral oscillations is associated with mild cognitive impairment [30]. Although we did not assess the cognitive state, a priori, our sample of women did not present any deterioration and therefore no significant oscillations were recorded in this front. It is safe to assume that with the changes that come with aging, by increasing the base of support and the angle of ambulation [31], older people could present greater oscillations in the mediolateral direction, which would need to be verified with other studies.

Our results show that there are no significant differences in the posturographic variables studied in situation 1 (ROA), 2 (ROC) and 3 (RGA), but a situation of clear instability in situation 4 (RGC) is marked with an increase in the three variables studied according to other studies carried out [32,33]. In fact, the values between ROC and RGA are very similar. We observed that, when two systems intervene together, balance remains stable with no significant increases in oscillations. Changes in the feet and eyes seem to be the cause of postural imbalances and their corrections are essential to achieve this balance [4]. In fact, the largest deficit found in the elderly who suffer repeated falls has been in the visual field [34], although these data refer to elderly people with a history of falling, unlike our sample studied.

In the RGC situation, only vestibular function is present, given the cancellation of the other two systems, with a progressive deterioration as age advances [35]. In effect, people with Menière’s disease are unable to maintain good stability in the RGC situation, but nevertheless they do have good visual or somatosensory information that improves their stability [36]. In the cases where improvement of vestibular or visual function were not possible, balance rehabilitation would be based on proprioceptive training as a preventative measure against falls. However, we must consider all the multiple causes that could interfere with the loss of balance individually, such as parameters related to muscle function (tone, elasticity, EMG activity) [37].

## 5. Conclusions

Both cancellation of visual and somatosensory afferents could facilitate the appearance of falls because this causes a situation of maximum instability, demonstrated by a significant increase in both body sway area and AP and ML displacements, with no correlation to age.

The posturographic platform confirmed that, although within normal parameters, the only percentage variations occurred in the somatosensory system, which could steer balance training towards somatosensory or proprioceptive improvement as a fundamental pillar in the prevention of falls.

The anteroposterior oscillations are greater than the mediolateral ones, which could be because the women in the study are a healthy population without cognitive impairment. It is necessary to identify each situation individually to establish appropriate preventive measures.

## Figures and Tables

**Figure 1 sensors-21-07684-f001:**
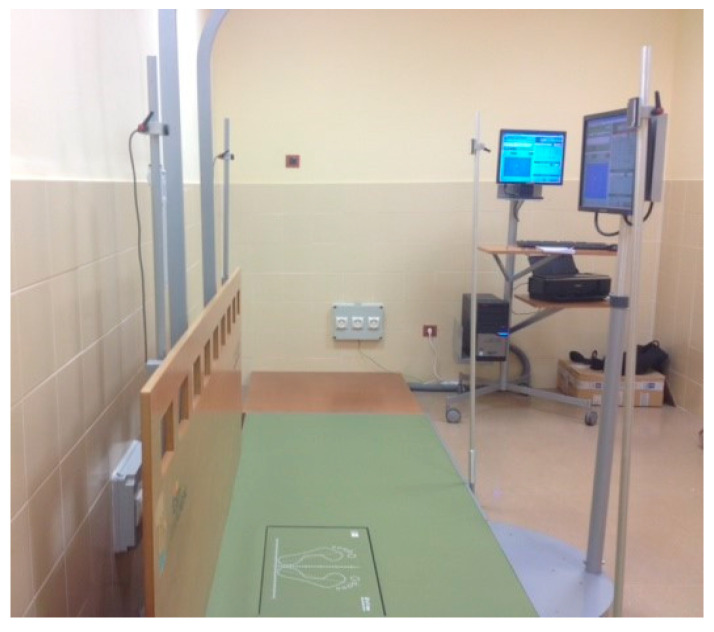
Dinascan/IBV force platform.

**Figure 2 sensors-21-07684-f002:**
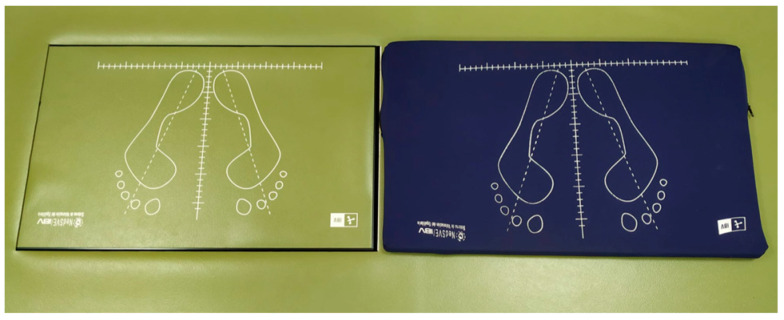
Foam rubber pad used in RGA and RGC.

**Table 1 sensors-21-07684-t001:** Anthropometric characteristics of participants.

	Mean ± SD	Minimum	Maximum
Age	62.84 ± 7.77	50	78
Weigh	65.07 ± 9.05	40.6	78.1
Heigh	157.94 ± 5.7	143	168
IMC	26.08 ± 3.57	19.68	35.73

**Table 2 sensors-21-07684-t002:** Changes and levels of significance (*p* = 0.05) in the variables according to the study situation. ROA: Romberg, eyes open; ROC: Romberg, eyes closed; RGA: Romberg, foam, eyes open; RGC: Romberg, foam, eyes closed.

	ROA	ROC	RGA	RGC	*p*
Body sway area (mm^2^)	29.81 ± 18.27	43.15 ± 28.64	48.70 ± 37.57	143.07 ± 115.35	0.001
AP displacement (mm)	18.15± 8.39	23.30 ± 10.03	22.27 ± 7.93	38.03 ± 14.89	0.001
ML displacement (mm)	10.06 ± 3.09	13.84 ± 6.71	15.60 ± 12.45	24.56 ± 13.31	0.001

**Table 3 sensors-21-07684-t003:** Correlation between age and variables in each situation. ** Pearson coefficient (r).

	ROA		ROC		RGA		RGC	
	r/rho	*p*	r/rho	*p*	rho	*p*	rho	*p*
Body sway area	−0.009	0.95	0.09	0.54	0.23	0.88	0.16	0.26
AP displacement	−0.09 **	0.54	0.27 **	0.85	−0.001	0.99	0.15	0.30
ML displacement	0.18	0.21	0.22	0.12	0.19	0.18	0.12	0.40

## Data Availability

Not applicable.

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
