# Peer review of "An Assessment of Balance through Posturography in Healthy about Women: An Observational Study"

_sensors, 2021, doi:10.3390/s21227684_

Round 1

Reviewer 1 Report

In this manuscript, Escamilla-Martínez and colleagues aimed to characterize a sample of older women (> 50 years old) with no previous history of falls from a biomechanical standpoint, namely the existence of changes of balance and its relationship with age. Overall the study is interesting and relevant. However, it feels somewhat underdeveloped, namely regarding the biomechanical characterization of the subjects. Many subject variables are missing - mean age +/- sd; weight, height, BMI; height of center of gravity and anthropometric measures. Also, the manuscript should be proofread by an English native speaker in order to improve the quality of the text, especially the syntax.

 Besides that, I would like to leave some comments/questions:

  1. Surely there are other medications that can affect balance besides the ones mentioned in the manuscript (i.e., diuretics, anti-arrhythmics,...). Did you consider them as well?
  2. Please mention how many subjects of a certain age you included (how many from 50 to 55 y.o., from 55 to 60 y.o., etc.);
  3. How did you determine the sample size?;
  4. Lines 86-87 - this sentence seems confusing. If the subjects were allowed to find their most stable position while standing barefoot, why were they asked to remain with their heels together and with their feet separated by a 30 degree angle? Please clarify;
  5. Figure 2 should be improved, especially in terms of sharpness;
  6. Figure 3 is irrelevant and should be removed as it does not contribute anything to the reading and interpretation of the study;

Author Response

Dear reviewer:

Thank you for giving us the opportunity to resubmit our work. We have revised the manuscript according to the suggestions and comments made you. We made a point-by-point response to the comments and edited our manuscript accordingly. We thank you in advance for taking time to consider our manuscript and eagerly await your response.

Yours faithfully,

Dr. Lourdes María Fernández-Seguín

Reviewer 2 Report

Very interesting study. I congratulate the authors for such an excellent study. Please see the attached file for some minor suggestions. Discussion must be improved.

Author Response

(The authors gave the same response as above.)

Round 2

Reviewer 1 Report

The authors have improved their manuscript according to the reviewers' suggestions. The manuscript has now sufficient quality for publication, although an English language revision is encouraged.